# Emerging Issues in Mapping Urban Impervious Surfaces Using High-Resolution Remote Sensing Images

**Zhenfeng Shao** [1], **Tao Cheng** [1,*], **Huyan Fu** [2], **Deren Li** [1] **and Xiao Huang** [3]

1  State Key Laboratory of Information Engineering in Surveying, Mapping and Remote Sensing,
   Wuhan University, Wuhan 430079, China
2  School of Earth Sciences, Yunnan University, Kunming 650500, China
3  Department of Geosciences, University of Arkansas, Fayetteville, AR 72701, USA
*  Correspondence: taocheng@whu.edu.cn

**Abstract:** Urban impervious surface (UIS) is a key parameter in climate change, environmental change, and sustainability. UIS extraction has been evolving rapidly in the past decades. However, high-resolution impervious surface mapping is a long-term need. There is an urgent requirement for impervious surface mapping from high-resolution remote sensing imagery. In this paper, we compare current extraction methods in terms of extraction units and extraction models and summarize their strengths and limitations. We discuss the challenges in impervious surface estimation from high spatial resolution remote sensing imagery in terms of selection of spatial resolution, spectral band, and extraction method. The uncertainties caused by clouds and snow, shadows, and vegetation occlusion are also analyzed. Automated sample labeling and remote sensing domain knowledge are the main directions in impervious surface extraction using deep learning methods. We should also focus on using continuous time series of high-resolution imagery and multi-source satellite imagery for dynamic monitoring of impervious surfaces.

**Keywords:** impervious surface estimation; urban mapping issues; remote sensing

## 1. Introduction

Urbanization contributes to changes in urban spatial structures and land surface properties [1]. These changes are primarily a process of conversion from natural land surfaces to urban impervious surfaces (UISs). UIS, a key environmental indicator in climate change, environmental change, and sustainability studies, has become a current research hot topic. UIS refers to a land surface paved with impervious or low permeability materials within the urban development boundary. UIS generally consists of materials such as tile, impervious asphalt, and impervious concrete. It typically includes buildings, structures, impervious roads, plazas, parking lots, etc. [2,3]. In the past few decades, many people have poured into cities [4], accelerating the urbanization process and leading to the rapid expansion of UIS. The high density of UIS has caused many urban problems, e.g., urban heat islands [5], urban waterlogging [6], soil erosion [7], and air pollution [8]. Therefore, the mapping UIS can be used to monitor urban expansion, population change, and environmental change [8,9].

To accurately analyze the current status of impervious surfaces research, we use the two groups of keywords, "impervious surface(s)" and "impervious surface(s) + high resolution", to analyze the number of articles in the Web of Science Core Collection from 2000 to 2022, with a total number of 757 and 109, respectively (see Figure 1). Figure 1 depicts the number of articles per year for keywords "impervious surface(s)" and "impervious surface(s) + high resolution", respectively, and plots the proportion of "impervious surface(s) + high resolution" articles to "impervious surface(s)" articles each year. We find that the number of impervious surface publications is showing an increasing trend, with a flat

trend between 2019 and 2022. Despite the presence of annual publications on impervious surfaces, studies on high-resolution remote sensing images are still rare.

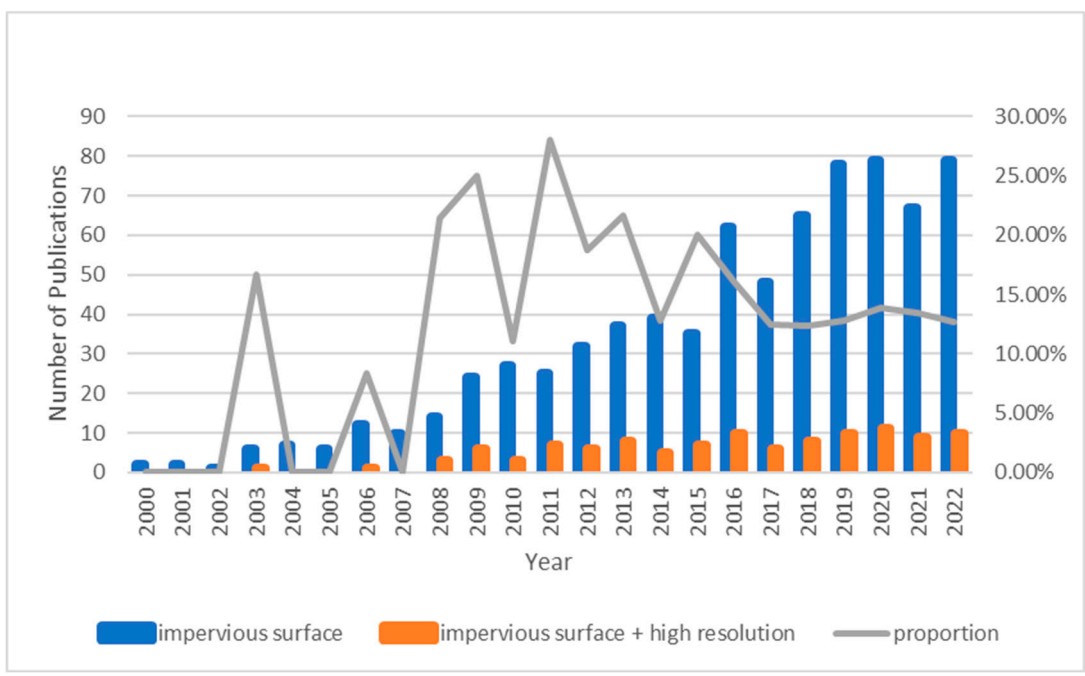

**Figure 1.** Literature statistics of urban impervious surface. The statistics are counted according to the two groups of keywords, "impervious surface(s)" and "impervious surface(s) + high resolution", from 2000 to 2022 in the Web of Science Core Collection, acquiring a total of 757 publications and 109 publications, respectively.

Due to the free and open data policy, advanced cloud computing platforms (e.g., Google Earth Engine (GEE), Pixel Information Expert Engine (PIE-Engine)) provide powerful computational capabilities and large amounts of online data for global-scale and regional-scale impervious surface studies. As a result, the number of publications on impervious surfaces has increased in recent years. With the improvement of the basic theoretical framework and the deepening of scientific research, low- and medium-resolution impervious surfaces can no longer meet the scientific problems at fine scales. It is urgent to extract high-resolution impervious surfaces quickly and accurately to explore and study key scientific questions at fine scales.

Although high-resolution optical remote sensing imagery and low- and medium-resolution optical remote sensing imagery have similar spectral characteristics, the differences in geometric and textural characteristics are enormous. As the spatial resolution increases, the data volume of high-resolution remote sensing imagery increases geometrically, requiring more parallel computing capabilities for impervious surface mapping. At the same time, the spectral differences between the same objects increases the difficulty of fine-grained impervious surface extraction. Some excellent reviews on impervious surfaces have been published [9–13], but there is a lack of reviews on impervious surfaces that focus only on high-resolution remote sensing imagery. Therefore, we discuss and analyze the requirements, methods, issues, and extraction strategies for the extraction of impervious surfaces from high-resolution remote sensing imagery in detail.

The content of this paper is structured in the following order. Section 2 elaborates on the requirements for mapping urban impervious surfaces using high-resolution remote sensing images. Section 3 analyzes various existing methods for extracting urban impervious surfaces from high-resolution remote sensing images. Section 4 discusses impervious surface extraction from high-resolution remote sensing images in terms of spatial resolution selection, spectral band selection, extraction method selection, and uncertainty. Finally,

Section 5 provides recommendations to improve our understanding of impervious surface monitoring, both its theoretical and practical aspects.

## 2. Requirement on Mapping Urban Impervious Surfaces Using High-Resolution Remote Sensing Images

### 2.1. Urban Surface Energy Balance

In the context of rapid urbanization, a higher proportion of impervious surfaces alters the heat capacity, albedo, and local climate conditions, affecting surface energy absorption, storage, and emittance [14–16]. Thus, the rapid growth of impervious surfaces changes the mode of surface energy exchange [17]. Due to high heat capacity and heat conductivity, urban land surfaces tend to absorb a large amount of solar radiation energy, leading to an increase in urban surface temperature, an acceleration of the hydrological cycle, and more extreme rainstorm events.

The change in the spatial structure of impervious surfaces is likely to impact surface heat distribution and aggravate thermal environment issues. In order to study the impact of urban change on the urban thermal environment, we can take impervious surfaces as the representation of urban change, build a theoretical framework between impervious surfaces and the urban thermal environment, and put forward reasonable suggestions to mitigate the urban heat problem.

Urbanization has a significant impact on the urban hydrological cycle mechanism. Impervious surface, as the most prominent feature of urbanization, leads to a massive and comprehensive change in the hydrological system across different spatial scales [18,19]. Due to the impermeability, an increase in impervious surfaces is likely to lessen evapotranspiration and infiltration, leading to increased stormwater runoff. With the decrease in infiltration, there is a direct reduction in vertical infiltration recharge to groundwater, leading to a decrease in groundwater levels. We argue that accurate, high-resolution mapping of impervious surfaces is conducive to the analysis of the urban hydrological cycle mechanism.

### 2.2. Sustainable Urban Development

At current population growth rates, 60% of the world's population will live in cities by 2030 and 68% will do so by 2050 [4] As the population migrates to cities, the amount of impervious surfaces (e.g., urban built-up lands) increases dramatically. We acknowledge the contradictions among the population, built-up lands, and ecology posing challenges to the sustainable development of cities. Therefore, better management of urban expansion and population growth paves the way for urban sustainability.

Among the 17 Sustainable Development Goals (SDGs) proposed by the United Nations in 2015 [20], the SDG11.3.1 indicator, which refers to the Ratio of Land Consumption Rate to Population Growth Rate (LCRPGR), can be used to quantify the coordination between urban land expansion and population growth. Studies have indicated that the impervious surfaces extracted from remote sensing images can accurately represent urban surface information [21,22]. High-resolution impervious surfaces can better present the urban internal structure and accurately map the spatial differences of urban sustainable development [23]. Therefore, it is urgent to obtain large-scale and high-resolution urban impervious surfaces spatial distribution information to monitor sustainable urban development.

### 2.3. Old City Reconstruction and New Urban Construction

Rebuilding the underground pipeline network for drainage is difficult and can lead to traffic congestion and extra excavation costs. Thus, it has caused a major demand to increase the permeability of surface features, such as turning impervious surfaces into permeable surfaces. The old city reconstruction and the new urban construction are expected to have an important influence on urban development. However, a series of challenges (e.g., urban waterlogging) are likely to be introduced.

In recent years, there has been a growing interest in investigating inland inundation using different models. Ecological models are currently being used to analyze urban waterlogging problems. The underlying principle of these models is that the amount of rain exceeds the discharge capacity of the urban drainage system due to the high percentage of impervious surface. Therefore, accurate information about urban impervious surfaces can lead to new solutions for urban waterlogging.

The extraction of urban impervious surfaces from remote sensing technology is beneficial for the old city reconstruction and new urban formulation, allowing us to create a "breathing" city, with the permeability of urban areas through the cavernous transformation of the old urban area. As for the formulation of the new urban areas, the permeability should also be considered, as the amount and distribution of impervious urban areas closely correlate with land use.

## 3. Methods

Traditional impervious surfaces are mainly characterized by spectral information from low spatial resolution satellite imagery rather than texture and local features. With the improvement of satellite sensors, high-spatial-resolution remote sensing images have become dominant. Clear boundaries, spatial structures, and texture features can be derived from high-resolution imagery to extract fine urban impervious surfaces. Various existing methods can be classified based on extraction units or involved models (see Figure 2).

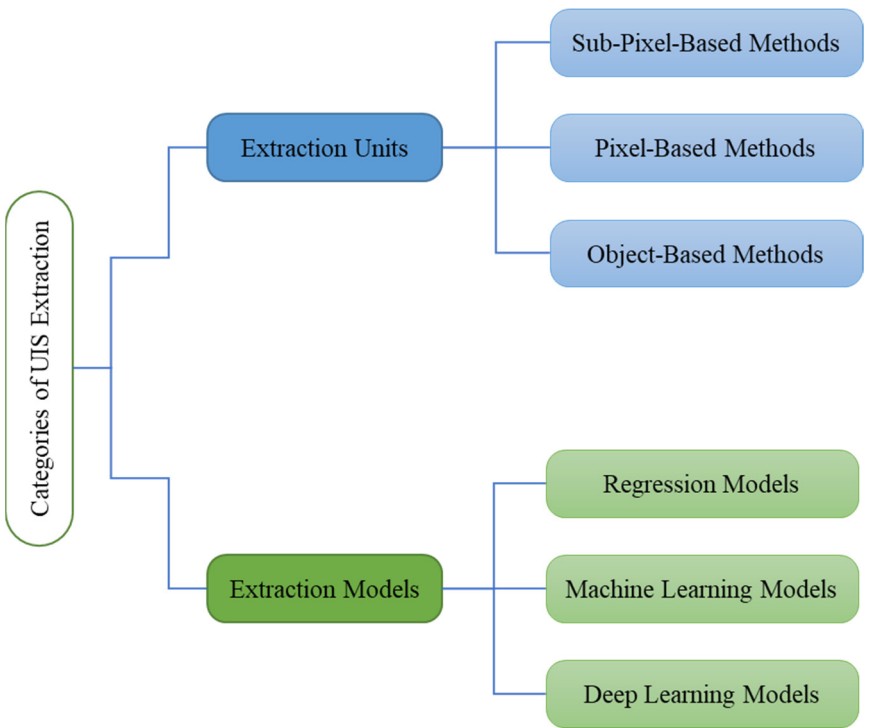

**Figure 2.** The generalized classification of major UIS extraction algorithms for high-spatial-resolution remote sensing images.

### 3.1. Methods Classified by Extraction Units

3.1.1. Sub-Pixel-Based Methods

At present, satellite remote sensing data at low/medium spatial resolution, such as that obtained via Landsat TM/ETM (Thematic Mapper/Enhanced Thematic Mapper), MODIS (Moderate-Resolution Imaging Spectroradiometer), Hyperion, AVHRR (Advanced Very High Resolution Radiometer), and DMSP/OLS (Defense Meteorological Satellite Program-Operational Linescan System) [24–32], have been mainly used to quantify sub-pixel impervious surfaces at global and regional scales [33,34]. The classification methods

mainly include the sub-pixel classifier, machine learning algorithm, and spectral mixture analysis [35–39]. The mixed pixel issue has been greatly reduced with the advent of high spatial resolution remote sensing images since the 1990s, e.g., IKONOS (launched 1999) and Quick Bird (2001). However, mixed pixels are still present at the edge of the land cover. Mohapatra and Wu [40] utilized Artificial Neural Networks (ANN) to extract sub-pixel impervious surfaces from high-resolution satellite imagery and showcased the great potential of sub-pixel methods in impervious surface extraction. Wu [41] explored the applicability of spectral mixture analysis (SMA) for impervious surface estimation using IKONOS imagery. This study shows that IKONOS imagery contained 40–50% of mixed urban pixels for the study area, and the within-class variability is a severe problem for the spectral analysis. However, extraction of impervious surfaces from high-resolution remote sensing images tends to ignore the mixed pixel problem. Therefore, few studies have been conducted to extract the sub-pixel impervious surfaces from high-resolution remote sensing images [42].

### 3.1.2. Pixel-Based Methods

Pixel-based methods are equivalent to a binary classification of features and mainly depend on traditional supervised and unsupervised classification methods, where high-resolution remote sensing images are often the suitable data resources. Popular pixel-based methods include ISODATA and maximum likelihood classification [43,44]. Hu and Weng [45] used IKONOS images to extract impervious surfaces for Indianapolis. Xu [46] developed a rule-based method to extract impervious surfaces with high-resolution imagery from IKONOS, ALOS, and SPOT-5 in three cities. The designed rules effectively distinguish impervious surface features from the land use types such as soil and water. Although pixel-based classification methods retain the details of the original image at the pixel level, classification results can be easily disturbed by various factors, such as solar radiation angle and soil moisture, leading to salt-and-pepper noises.

### 3.1.3. Object-Based Methods

Object-based methods have received increasing attention in urban impervious surface extraction from high-resolution remote sensing imagery [47–51]. For object-based methods, various features can serve as the model input, such as spectral information, texture features, spatial information of objects, shape features, and characteristics of proximity relationships. Furthermore, fuzzy logic rules are often introduced in object-based methods to reduce confusion among different surface features, thereby improving the accuracy of impervious surface extraction [52].

Lu et al. [53] compared pixel-based, segmentation-based, and hybrid methods for mapping impervious surfaces using high spatial resolution data in Brazil's urban landscapes. The hybrid method provides the best performance with reduced "salt-and-pepper" issues. However, it requires considerable time and labor, involving manual editing and refining impervious surfaces. Berger et al. [54] employed the object-based image analysis (OBIA) approach to estimate impervious surfaces from high-resolution multi-spectral optical imagery and LiDAR data. Based on prior knowledge and a weighted minimum distance strategy, Zhang et al. [50] proposed a pixel- and object-based hybrid analysis (POHA) method, which could provide accurate impervious surface mapping with limited human–computer interactions. Jebur et al. [55] used three types of classifiers (i.e., Support Vector Machine (SVM) pixel-based, SVM object-based, and Decision Tree (DT) pixel-based classification) for mapping impervious surfaces. Their study reveals that object-based SVM is better than pixel-based SVM, both superior to the DT model. Image segmentation is vital to the object-oriented classification method. Nonetheless, there are difficulties in selecting the best segmentation parameters and methods that pose challenges. In addition, image segmentation is affected by factors such as light, noise, and shadow [56].

The mixed pixel issue often occurs in medium- and low-resolution satellite images. High-resolution remote sensing imagery can greatly mitigate this issue. However, many

problems deserve our attention, such as topographical variation and shadows cast by tall buildings and canopies. At the same time, due to the limited spectral bands onboard the high-resolution remote sensing sensor, the phenomenon of "different things having the same spectrum" is common, greatly limiting the sensors' capability in differentiating impervious surfaces from other land cover types from a spectral perspective. To deal with this problem, we have two options, one is to add additional data sources [57], and the other is to extract more complex features. Without additional data sources, spectral confusion can be reduced by more comprehensive features (combined with spectral, spatial, and texture features) [11,58]. Table 1 provides a summary of the strengths and limitations of these extraction units, with the relevant literature.

**Table 1.** Summary of the strengths and limitations for these extraction units.

| Extraction Units | Advantages | Disadvantages |
| --- | --- | --- |
| Sub-pixel-based | physical meaning; proportion of endmember | uncertain endmember number; large differences between pure pixels of the same object |
| Pixel-based | data parallel; fast computation | fragmented classification results |
| Object-based | geometric and spatial features; non-fragmented classification results | slow segmentation |

### *3.2. Methods Classified by Extraction Models*

#### 3.2.1. Regression Models

A regression model is a mathematical tool to quantitatively describe the statistical relationship which aims to seek the feature variable(s) that highly correlate with impervious surfaces. The classification and regression tree (CRAT) is sensitive to noise and training sample error [59]. Compared to the single CART algorithm, the integrated CART often presents better stability and robustness. Jiang et al. [60] used the CART algorithm to characterize impervious surfaces in Hong Kong by integrating the SPOT-5 multispectral image and ERS-2 SAR data. Jiang et al. [60] used regression analysis to extract urban impervious surfaces by fusing the SPOT data with InSAR data. It is easy to derive impervious surfaces using regression analysis in a short span of time [61,62]. However, such a method is likely to over/underestimate impervious surfaces due to seasonal variations [63].

#### 3.2.2. Machine Learning Models

Due to the complexity of urban terrain, the assumption of the normal distribution of land use and land cover cannot be achieved. As a result, some machine learning-based classifiers, such as Artificial Neural Networks (ANN), Support Vector Machine (SVM), and Random Forest (RF) [59,64–66], are considered more suitable for impervious surface estimation. These models can extract impervious surfaces from high-resolution remote sensing images at the pixel level and low spatial resolution remote sensing images at the object level. In some studies, these models can be combined with sub-pixel classification methods to retrieve fine-grained impervious surfaces [32,65].

(1)　Artificial Neural Networks

Commonly used ANN classifiers include MLP (Multi-Layer Perceptron), Hopfield Neural Network, ARTMAP, and SOM (Self-Organizing Map). MLP and SOM are the most widely used for mapping impervious surfaces. Compared with traditional classifiers, ANN is a nonparametric classifier that requires only a small number of samples to handle the nonlinear patterns without normality assumption, which can fuse auxiliary data and prior knowledge at a high fault tolerance [67,68]. Im et al. [59] proposed artificial immune networks and decision/regression trees in quantifying impervious surfaces by integrating the high spatial resolution WorldView-2 image with the LiDAR data.

The ANN classifier is highly dependent on the quantity and quality of the learning samples and tends to have a slow convergence speed and poor stability. For example, in MLP models, it is difficult to determine the number of hidden layers and the number of nodes in each hidden layer. In comparison, the classification accuracy of the SOM depends on the number of features. Too many (or too few) neurons are likely to affect impervious surface identification [69,70].

(2)　Support Vector Machine

Different from those traditional algorithms based on the empirical risk minimization rule, SVM is based on the structural risk minimization rule. Thus, it can achieve a great balance between empirical risk and classifier capacity. Jebur et al. [55] used three types of classifiers (SVM pixel-based, SVM object-based, and Decision Tree (DT) pixel-based classification) in characterizing impervious surfaces from SPOT 5 images. Their study shows that the object-based SVM is better than the pixel-based SVM, both superior to the DT in quantifying impervious surfaces. Leinenkugel et al. [71] mapped the impervious surfaces using SVM based on the object-oriented model that combined the TerraSAR-X and SPOT-5 images. Foody et al. [72] compared SVM, DT, and MLP methods in mapping impervious surfaces and concluded that the SVM technique is better than the DT and MLP ones. The SVM method can achieve satisfactory performance with a small number of training samples, but these samples are difficult to locate.

(3)　Random Forest

The Random Forest technique is originally proposed based on the Decision Tree classification model by Breiman [73]. Random Forest (RF) has been widely used in remote sensing image classification due to its ability to mitigate overfitting and deal with noise and high dimensionality in the dataset [74–78]. Shao et al. [79] used RF to generate an accurate urban impervious surfaces map from GaoFen-1 (GF-1) and Sentinel-1A imagery. Multi-source and multi-sensor remote sensing datasets are combined to estimate impervious surfaces [64]. The built-in out-of-bag (OOB) error is insufficient for accuracy assessment and additional reference data is required for quantifying impervious surfaces [80]. The RF method is easy to parallelize and insensitive to the spatial resolution and sensor type.

### 3.2.3. Deep Learning Models

Due to the complexity of urban surfaces in high-resolution remote sensing images, the utilization of shallow machine learning methods (e.g., CART, ANN, SVM, and RF) with limited samples may fail to present satisfactory results [81]. Another issue is that models trained by the shallow machine learning method often fail to be transferred to other study areas. In comparison, convolution neural networks (CNNs) have the capability to automatically learn features from massive remote sensing images, with strong generalization capability.

Since CNNs achieve excellent results in the ImageNet Large Scale Visual Recognition Challenge 2012 (ILSVRC2012) [82], they have been widely used in remote sensing image classification [83–87]. The CNN method achieves higher accuracy in the extraction of urban impervious surfaces than the traditional method [81]. Although CNNs have been adopted for feature extraction, many land covers cannot be correctly identified with limited spectral bands. Therefore, multi-source remote sensing data fusion has been explored by numerous scholars to further improve classification accuracy.

The workflow, depicted in Figure 3, consists of two complementary parts. In the first part, a remote sensing image is fed into a deep convolution network with de-convolution layers for discriminative feature learning. Then, a softmax classifier is used to predict the class probability for each pixel. In the second part, the remote sensing image is regarded as a graph, and a conditional random field (CRF) model with spatial consistency constraints is implemented for global label optimization.

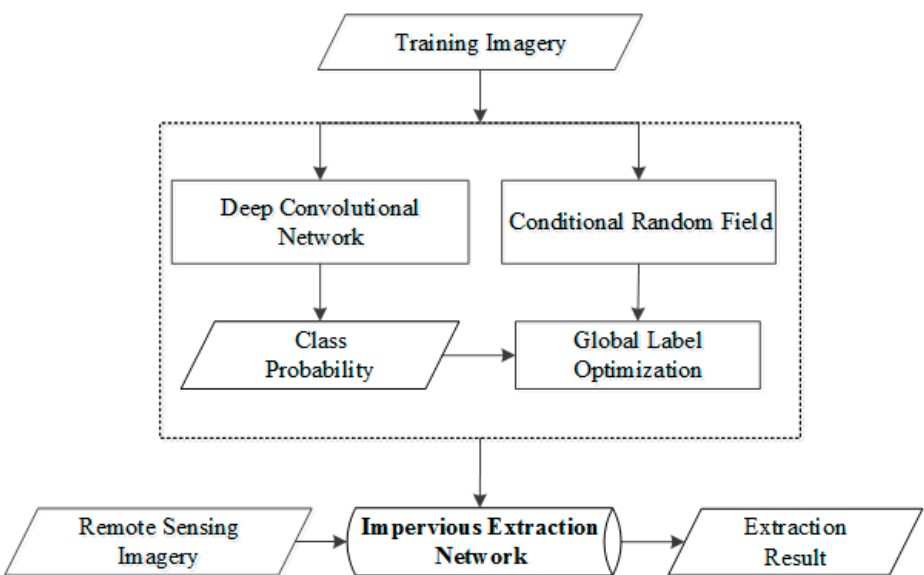

**Figure 3.** Diagram of the impervious surface extraction.

(1)    Class probability prediction using the deep convolution network

The deep convolution network for impervious extraction is shown in Figure 4. The network architecture consists of two main parts: a feature learning part and a classification part. The feature learning part is composed of standard convolutional neural networks (CNNs) with convolution layers and pooling layers. The classification part is composed of deconvolution layers and a softmax classifier. The output of the network is the predicted class probability for each pixel of the input image.

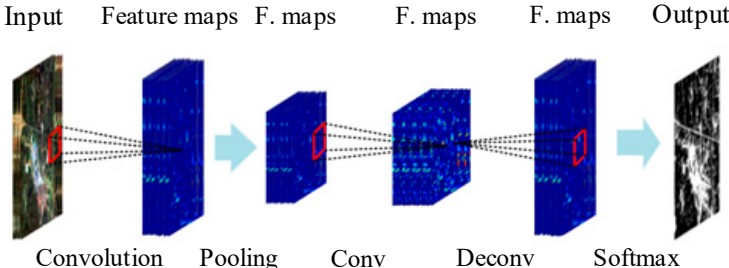

**Figure 4.** The deep convolution network for extracting impervious surface.

(2)    Global label optimization using the CRF

The predicted class probability at the pixel level is inevitably influenced by the data noise, and the spatial consistency is hardly considered. To address this issue, a conditional random field model with spatial consistency is used to globally optimize the label of the whole image.

The input image is regarded as a graph $G(V, E)$ with vertex $v \in V$ and edge $e \in E$. Each vertex is associated with a pixel, and edges are added between the pixel and its K-nearest neighbors. The CRF energy function is typically composed of a unary term enforcing the variable $l$ to take values close to the predictions $\hat{p}$ and a pairwise term enforcing regularity or local consistency of $l$. The data cost term $\varphi(\hat{p}_i, l_i)$ is used to penalize the disagreement between a point and its assigned label. The initial data cost of each point is calculated with its predicted probability.

$$\varphi(\hat{p}_{i,k}, l_i) = \exp(-\hat{p}_{i,k})\mathbf{1}(l_i \neq k),\tag{1}$$

where $\hat{p}_{i,k}$ corresponds to the probability that a pixel $i$ belongs to class $k$.

The label inconsistency between neighboring points is penalized by the smooth cost term $\psi(l_i, l_j)$. The neighboring points are encouraged to have similar labels. The smooth cost is calculated according to the gradient of the image, i.e.,

$$\psi(l_i, l_j) = \exp(-\|\nabla I\|_i)\mathbf{1}(l_i \neq l_j),\tag{2}$$

where $\nabla I$ is the gradient of the input image $I$. The smooth cost constrains the regularity and consistency of label $l$.

(3) Experimental result

The high-resolution image that covers a small part of Wuhan is used to extract impervious surfaces. The size of the image is $862 \times 837$ pixels. The image is first classified into six land cover types, including building, road, water, vegetation, shadow, and bare. Figure 5b shows the spatial distributions of impervious surfaces. There are still misclassifications between water and shadow in Figure 5c. A specific network for water and shadow classification facilitates highly accurate impervious surface extraction. Various types of buildings make it difficult for extracting buildings accurately. Pixel-wise extraction and global optimization can hardly guarantee the geometric shape of objects. Table 2 provides a summary of the strengths and limitations of these extraction models, with the relevant literature.

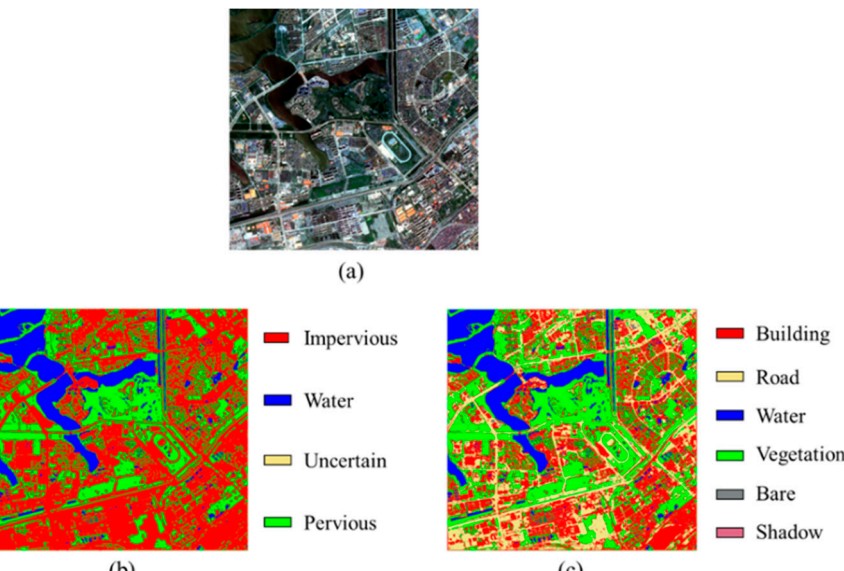

**Figure 5.** The impervious surface extraction by the CNN method. (**a**) The original image; (**b**) The impervious surface; (**c**) Land cover patterns.

**Table 2.** Summary of the strengths and limitations for these extraction models.

| Method | Advantages | Disadvantages |
|---|---|---|
| Regression models | simple; rule-based modeling | sensitive to noises and training sample errors; over/underestimate impervious surfaces |
| Machine learning models | data parallel (RF); easy to work; simple parameters | dependent on data features |
| Deep learning models | self-learning; high-level semantic feature; data parallel; fast computation; easy-to-use GPU | black-box working method; need a large number of samples; difficulty in model training |

## 4. Discussion on Extraction Strategies

With the advent of high spatial resolution images, the spatial information of objects has been gradually introduced into impervious surface extraction. Lu et al. [56] discussed the scale issues and the section of data source for impervious surface extraction. According to their findings, high spatial resolution remote sensing images are suitable for extracting

texture features for mapping impervious surfaces. However, the limited spectral bands and the complex land cover pose great challenges to accurately extracting impervious surfaces. This also places higher demands on the method of impervious surface extraction.

### 4.1. Spatial Resolution Selection

High-resolution images can capture details and spatial relationships among different objects, gradually becoming an important data source for fine-grained urban impervious surface extraction. However, due to the complex urban settings, high spatial resolution images bring other issues. Taking the object-oriented method as an example, the spatial resolution of satellite images directly affects the image segmentation scale and the efficiency of segmentation. A higher spatial resolution may result in finer segmentation objects but a lower efficiency of the segmentation algorithm. A lower spatial resolution image may not be able to capture road details and some sparse objects. It was determined that images at 2 m spatial resolution (e.g., from GF-1, ZY-3, and TH) are appropriate for extracting urban impervious surfaces using the object-oriented method (see Figures 6 and 7). The results of image segmentation and accuracy of extraction become unsatisfactory when the image spatial resolution is at 4–8 m. When the spatial resolution of an image is less than 2 m, the rich spatial detail makes it more challenging to identify land cover types. As shown in Figure 6, the differences in objects on remote sensing images at different resolutions are dramatic, so it is necessary to choose an appropriate spatial resolution to meet the accuracy requirements and reduce the complexity of extracting impervious surfaces.

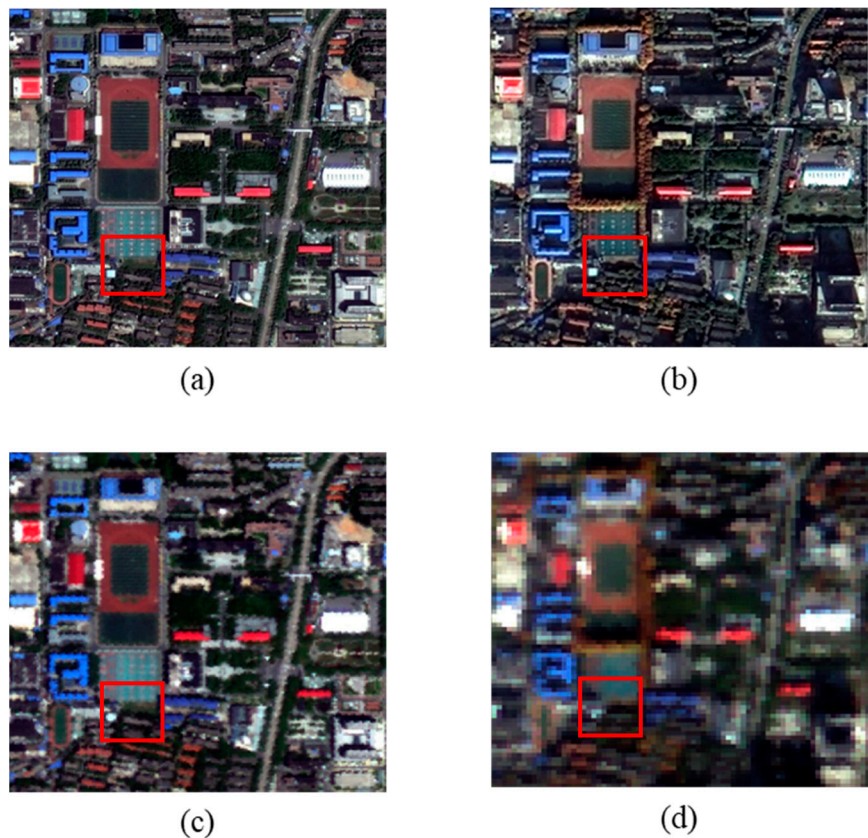

**Figure 6.** A comparison of remote sensing images with different spatial resolutions at the same geographical location. (**a**) Multispectral bands fused with panchromatic band in Gaofen-2 (0.8 m), 18 May 2020; (**b**) Multispectral bands fused with panchromatic band in Gaofen-1 (2 m), 11 December 2019; (**c**) Multispectral bands of Gaofen-2 (4 m), 18 May 2020; and (**d**) Multispectral bands of Gaofen-1 (8 m), 11 December 2019.

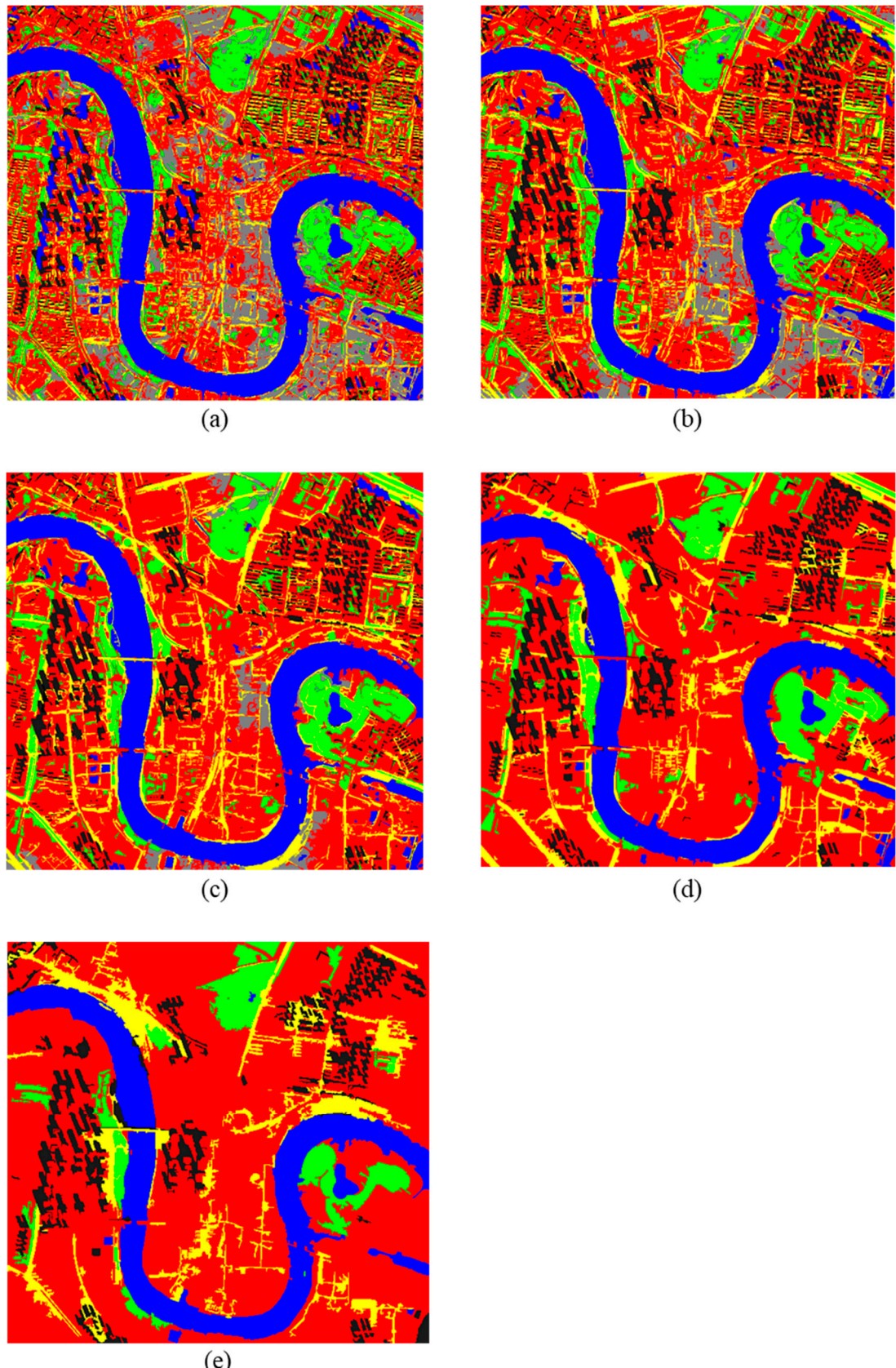

**Figure 7.** Impervious surface extraction by object-oriented methods at different spatial resolutions: (**a**) 0.5 m, (**b**) 1 m, (**c**) 2 m, (**d**) 4 m, (**e**) 8 m, shadow areas (black), impervious surface (red and yellow), water (blue), vegetation (green), bare soil (gray).

### 4.2. Spectral Band Selection

Accurately extracting impervious surfaces from high-resolution remote sensing imagery with only three visible bands remains challenging. As shown in Figure 8, water, vegetation and other land covers can be further identified by introducing the near-infrared (NIR) band and various indices, e.g., NDVI and NDWI. The near-infrared band can greatly improve the classification accuracy of water, vegetation, and shadow. The construction of a sponge city increases the number of low-impact development facilities, which turns the previous UIS into a permeable surface. Therefore, identifying the land cover for this area requires the use of high-resolution hyperspectral imagery. However, the correlation between adjacent bands of hyperspectral imagery becomes relatively high, leading to the issue of spectral redundancy. Band selection is particularly important to reduce spectral redundancy and accelerate image processing and classification.

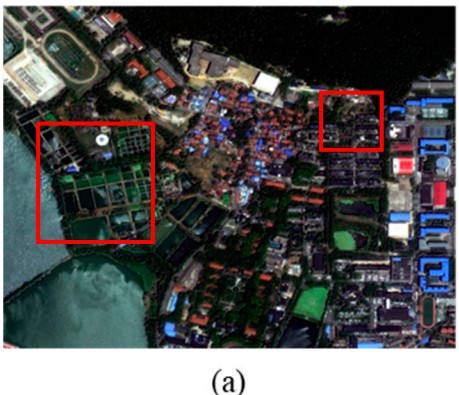 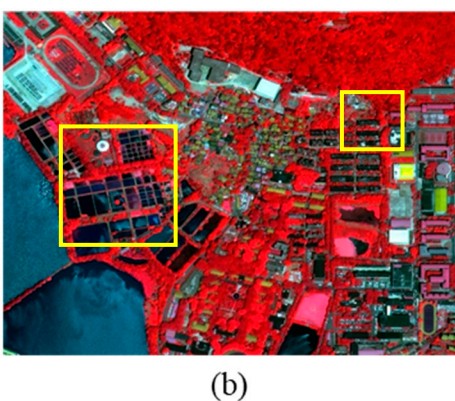

(a)      (b)

**Figure 8.** A comparison of remote sensing images in the visible and near-infrared bands. (**a**) True-color composites of Gaofen-2 images; (**b**) False-color composites of Gaofen-2 images.

### 4.3. Extraction Method Selection

One of the challenges of high-resolution remote sensing imagery is the very large amount of data. The training and inference of the model need to support data parallelization and computation parallelization. With the support of cloud computing platforms such as GEE and PIE-Engine, storage resources and computational resources can be guaranteed. At the same time, we need algorithms that support parallelization, such as DT, RF, and CNNs, to accelerate model training and inference. RF is the most widely used method in global impervious surface product mapping [10]. However, CNNs are self-learning and can learn high-level semantic features, which have great potential. The disadvantage is that CNNs require a large amount of sample data. Therefore, we should choose the appropriate method according to the difficulty of sample collection.

### 4.4. Uncertainty

High-resolution multispectral remote sensing imagery is the primary data source for extracting high-resolution urban impervious surfaces. The complex urban land cover brings a huge uncertainty. Many remote sensing images suffer from three problems due to sensor limitations: cloud and snow contamination, shadows, and vegetation cover. The easiest way to address these issues is to introduce homogeneous data for data fusion. However, considering the difficulty of data acquisition, heterogeneous data are often used as a supplementary data source for decision-level fusion.

#### 4.4.1. Cloud and Snow Contaminations

Optical images are inevitably contaminated by clouds, snow, cloud shadow, and other poor atmospheric conditions. For the very same surface object, spectral reflectance can present variance under clear-sky and haze/thin-cloud conditions. In most cases, it is still difficult to distinguish certain land types covered by snow and ice. Thus, additional

information is required to improve the mapping accuracy of impervious surfaces. SAR data can provide complementary information for optical images due to its all-day and all-weather capabilities at high spatial resolution and low cost [88–90]. Zhang et al. [80] combined multi-source and multi-sensor remote sensing datasets (i.e., Landsat ETM+, SPOT-5 and ASAR and ALOS PALSAR, SPOT-5, and TerraSAR-X) to estimate impervious surfaces in Pearl River Delta. Their research shows that combining SAR and optical images improves impervious surface recognition by reducing the misclassification from asphalt, bare soil, shadow, and water.

### 4.4.2. Shadow

High-resolution remote sensing images can provide clear boundaries and abundant information for extracting fine urban impervious surfaces. It is hard to identify land cover in the shadow areas due to the low digital number (DN) value. Challenges still remain when detecting land cover types in shadow areas. LiDAR, as an important remote sensing data source, can capture ground surface elevation values and rich geometric features. Im et al. [59] proposed a method to quantify the impervious surface using artificial immune networks and decision/regression trees by integrating high spatial resolution WorldView-2 imagery with LiDAR data. Hodgson et al. [91] employed different algorithms, such as maximum likelihood, ISODATA, and rule-based algorithms, to extract impervious surfaces from the natural color aerial photography and LiDAR data. The involvement of LiDAR data significantly improves the estimation accuracy of impervious surfaces [92–94]. For shadow areas cast by vertical urban structures, a possible solution is to use the unmanned aerial vehicle (UAV) and street view to assist in extracting impervious surfaces.

### 4.4.3. Vegetation Cover

Impervious surfaces covered by tall tree crowns in the urban area (e.g., the vegetation on both sides of the road) cannot be identified from high-resolution remote sensing images. One of the possible solutions is to take advantage of the low-altitude UAV and street view data or OpenStreetMap (OSM). A spatio-temporal-spectral-angular observation model [95] is proposed, which integrates observations from UAV and mobile mapping vehicle platforms to identify precise impervious surface boundaries. The OSM road network can be matched and corrected with the street trees in the high-resolution remote sensing imagery, e.g., morphological feature-oriented algorithm [96], which successfully eliminates the obscuring effects and mitigates the underestimation of impervious surfaces. As shown in Figure 9, the location of impervious surfaces identified by OSM is not very accurate in some areas. We should pay more attention to the direct registration of OSM and image.

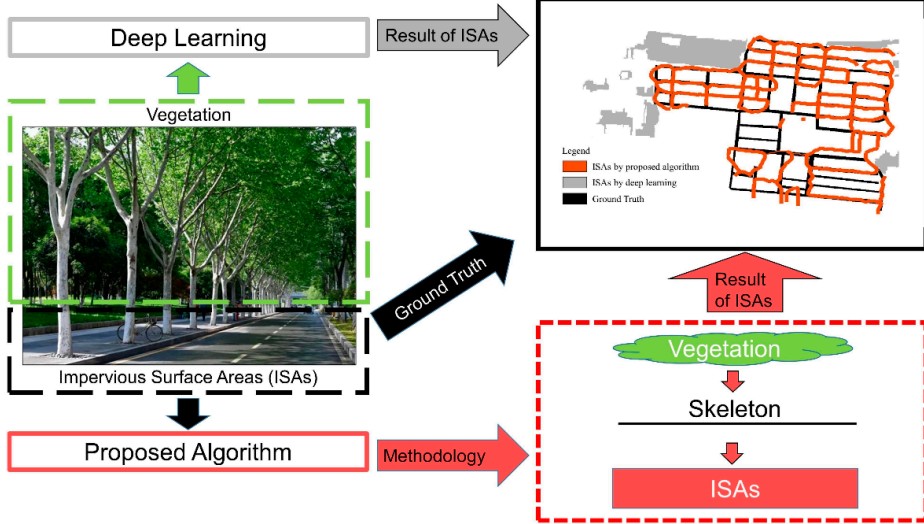

**Figure 9.** OSM-based impervious surface extraction under street tree occlusion [96].

*4.5. Future Prospects*

Deep learning models have great potential for high accuracy and automated mapping. Now, semantic segmentation network is developing towards the direction of weak supervision, light weight, and semantic reasoning [97,98]. To mitigate the issue of inaccurate boundary in prediction results, a semi-supervised semantic segmentation method based on boundary awareness [99] and a deep relearning method are proposed [100]. To overcome the problem of insufficient samples, Generative Adversarial Network (GAN) [101] and unsupervised convolution feature fusion network [102] are proposed. Graph neural networks are also proposed to deal with causal reasoning [103,104]. Thus, two main directions including automated sample labeling and the introduction of remote sensing domain knowledge should be emphasized.

**5. Conclusions**

Accurate identification of urban impervious surfaces is essential for a variety of applications, including hydrology, water quality, local climate, and biodiversity. Dynamic monitoring of impervious surfaces is an emerging perspective that can help us to understand urban land use/land cover dynamics and changes in urban ecological environments.

This study summarizes recent advances in urban impervious surface extraction using high-resolution remote sensing imagery. New algorithms, e.g., sub-pixel unmixing and fuzzy logic rules, have been developed to improve the identification of fine-grained impervious surfaces. However, the challenges associated with high-resolution imagery, such as shadows and noise, need to be mitigated and addressed. In the recent literature, high-resolution imagery has been widely used to map urban impervious surfaces, resulting in a number of unresolved issues. We provide the following recommendations to improve our understanding of impervious surface monitoring from both theoretical and practical aspects.

(1)  Dynamic monitoring of impervious surfaces using continuous time series of high-resolution images

There are many studies on the long-term identification of impervious surfaces, most of which focus on coarse scales. Challenges remain in dealing with mixed pixel issues, training sample selection, and classification assessment using low- or medium-resolution imagery. Moreover, heterogeneity within urban regions must be considered to properly manage the impacts of urbanization, such as stormwater mitigation and temperature regulation. High-resolution imagery provides spatial detail that captures the fine-scale heterogeneity within a metropolitan area and reduces the effects of mixed pixels. Therefore, we encourage more research on long-term monitoring of impervious surface dynamics using high-resolution imagery. However, it should be noted that higher-resolution data do not necessarily lead to a more accurate estimation. A synthetic view needs to be conducted considering the scale variations. In addition, the issues of identifying impervious surfaces covered by shadows and vegetation should be addressed in future research.

(2)  Dynamic monitoring of impervious surfaces using multi-source satellite images

Data fusion (or integration) of multi-sensor or multi-resolution data takes advantage of different data sources which potentially improves visual interpretation and quantitative analysis. The use of appropriate data fusion techniques is recommended to enhance the distinction between impervious surfaces and other land covers. The fusion of high- and medium-resolution data is an important research direction. For example, the regression relationship between predictive variables obtained from high-resolution data and feature variables extracted from medium-resolution multispectral data can be developed, followed by impervious surface identification using methods such as CART, ANN, and CNN. Integrating LiDAR-derived height information and high spatial resolution optical imagery is another important research aspect to improve the performance of impervious surface mapping. The combination of optical data, such as Landsat TM imagery and RADAR, is also beneficial for impervious surface mapping. For instance, both the texture and spectral



features of optical and SAR imagery can be combined to better distinguish impervious surfaces from other land covers. Coarse spatial resolution images, such as MODIS and Landsat, are available for large-scale time-series data and thus have become the primary data source for global impervious surface mapping. Therefore, more studies are needed to explore the possibility of integrating multi-scale and multi-sensor images.

**Author Contributions:** Conceptualization, T.C., H.F. and Z.S.; investigation, T.C.; writing—original draft preparation, T.C. and H.F.; writing—review and editing, T.C., D.L. and X.H.; visualization, T.C. All authors have read and agreed to the published version of the manuscript.

**Funding:** This research was supported in part by the National Natural Science Foundation of China under Grants 42090012, in part by the Guangxi science and technology program (Guangxi key R & D plan, GuiKe 2021AB30019); 03 special research and 5G project of Jiangxi Province in China (20212ABC03A09); Zhuhai industry university research cooperation project of China (ZH22017001210098PWC); Sichuan Science and Technology Program (2022YFN0031); Hubei key R & D plan (2022BAA048); Zhizhuo Research Fund on Spatial-Temporal Artificial Intelligence (Grant No. ZZJJ202202); and Guangzhou Basic and Applied Basic Research Project (202102020380).

**Data Availability Statement:** Not applicable.

**Acknowledgments:** We are grateful to those who participated in the manuscript revisions.

**Conflicts of Interest:** The authors declare no conflict of interest.

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
