# Peer review of "Emerging Issues in Mapping Urban Impervious Surfaces Using High-Resolution Remote Sensing Images"

_remotesensing, doi:10.3390/rs15102562_

Round 1

Reviewer 1 Report

Dear authors, the manuscript is informative but it has several inadequacies.  I shall highlight them and the authors might improve the quality and readability of this research paper accordingly. 

1. The introduction section does not provide a succinct theoretical basis for the study.  I would like to ask the authors to expand and highlight the advantages of their approaches that bring benefits in the solved issues.   

2. If possible, it will be good if the authors could add a graphical representation summarizing their results which compares controls, results, all the parameters and variables directly related to Urban Mapping Models. 

3. Please, briefly add future perspectives and further applied applications of this specific research work in the discussion section.  

4. The techniques and/or models presented and mentioned in the manuscript require sufficient details (including calibration, sensitivity analysis and validation) to allow other researchers to develop and test the applications later on.  Please include the parameters that I have mentioned here. More comparisons that show the advantages and the drawbacks of the proposed schema are needed.  

5. The most relevant data-results should be summarized and demonstrated by a graph and a corresponding table.  

6. Please, highlight the outliers in all the tables and graphs, where relevant.   

7. Please improve the quality of English Language in the manuscript. 

8. Please include and detail all the algorithms (mathematical expressions), of the related techniques and/or model/s mentioned in the manuscript.  

10. Please include a geo-referenced map of the study area shown within a world map.

11.The authors should clearly demonstrate how the applicability of the proposed method is better in comparison to other standard methods and how the proposed method opens up new avenues of quantitative research on impervious surface estimation and urban mapping issues deploying various types of remote sensing methods. 

12. The paper has been structured more like a review article. In its present form it does not appear to be a research manuscript. Please add the research components in order to publish the paper in a journal. 

Reviewer 2 Report

Dear Authors;

You have presented a good review paper and discussed the previous articles with dividing it with respect to related branches.

Best regards

Reviewer 3 Report

The mapping of urban impervious surface using high-resolution remote sensing imagery is reviewed in this manuscript along with the progress, issues, and prospects for future research. This work will help to understand the technological advances in the more frequent use of high-resolution imagery in urban studies, and is useful for ecologists aiming to use remote sensing to explore urban water and thermal issues. Therefore, even if the manuscript still needs revision, it is appropriate for publication in Remote Sensing after careful revision. Authors should consider the following suggestions and comments for manuscript revisions.

Comments and suggestions:

1. The paper's first five parts mostly list current studies, while the authors' opinions and comments on each are either absent or insufficiently detailed. I advise the authors to learn how to update this manuscript from every aspect of the review study Zhu et al. released in 2022.

 (https://doi.org/10.1016/j.rse.2022.113266)

2. Tools including time bars, figures, and tables are needed to show the progress of existing research more clearly, which also helps the authors to sort out the limitations of existing works to provide more in-depth insights.

 3. In Section 4, the authors mention three issues. An ongoing issue is how to classify impervious surfaces when clouds, shadows, and vegetations are present. Nonetheless, the spatial, temporal, and spectral resolution of the image plus the sensor's physical position are the primary drivers of these issues. Issues like classification methodology and computational power should also be taken into account. The authors may consider integrating the section IV and section V statements.

 4. The authors stress the effect of impervious surfaces on urban water issues in the introduction. This, however, is unrelated to the statements in the following sections in this paper's context. In other words, the introduction does not provide a strong foundation for the scientific aims of this study.

 5. The reference format should conform to journal standards.

Round 2

Reviewer 3 Report

Please  expand the abstract on the pivotal conclusions and new findings. The analysis framework need be improved next. 
